# Control of EMI in High-Technology Nano Fab by Exploitation Power Transmission Method with Ideal Permutation

**Yu-Lin Song** [1,2,*] , **Manoj Kumar Reddy** [1] , **Hung-Yi Lin** [3] **and Luh-Maan Chang** [3]

1   Department of Computer Science and Information Engineering, Asia University, Taichung 41354, Taiwan; manoj14a0kumar@gmail.com
2   Department of Bioinformatics and Medical Engineering, Asia University, Taichung 41354, Taiwan
3   Department of Civil Engineering, National Taiwan University, Taipei 10617, Taiwan; hungyilin@ntu.edu.tw (H.-Y.L.); luhchang@ntu.edu.tw (L.-M.C.)
*   Correspondence: d87222007@ntu.edu.tw

**Abstract:** There are many high-power electrical cables around and within semiconductor foundries. These cables are the source of extremely low-frequency (ELF < 300 Hz) magnetic fields that affect the tools which operate by the function of electronic beams. Miss operation (MO) happens because the ELF magnetic fields induce beam shift during the measurement or process for cutting-edge chips below 40 nm. We present the optimal permutation of power transmission lines to reduce electromagnetic influence in high-technology nano fabs. In this study, the magnetic field was reduced using a mirror array power cable system, and simulation results predicted the best permutations to decrease the electromagnetic interference (EMI) value to below 0.4 mG in a working space without any shielding. Furthermore, this innovative method will lower the cost of high-technology nano fabs, especially for the 28 nm process. The motivation behind this paper is to find the ideal permutation of power transmission lines with a three-phase, four-cable framework to decrease the EMI in high-technology nano fabs. In this study, the electromagnetic interference was diminished using the ideal-permutation methodology without investing or using additional energy, labor, or apparatus. Moreover, this advanced methodology will help increase the effectiveness and reduce the costs of nano fabs. The mathematical and experimental results of the study are presented with analysis.

**Keywords:** ideal permutation; electromagnetic interference (EMI)

## 1. Introduction

Over the past few years, the construction of nano fab industries has gradually increased day-to-day across the globe. While constructing a nano fab, one must focus on decreasing the electromagnetic interference (EMI) value exclusively for international research laboratories and independent organizations. In semiconductor foundries, there are many electrical high-power cables. These cables are the basis for extremely low-frequency magnetic fields (ELF < 300 Hz) that disturb apparatuses functioning by electronic beam operation [1,2].

Some of these tools include electron microscopes (e.g., scanning electron microscopy (SEM), which is not only used for nanomaterials characterization but also in the latest in-situ nanomaterials engineering technology), scanning transmission electron microscopy (STEM), and electron backscatter diffraction (EBSD) [3,4]. In high-technology nano fabs, while manufacturing cutting-edge chips below 40 nm, miss operation (MO) occurs due to ELF magnetic fields in the beam shift during the measurement. To avoid MO, the International Technology Roadmap for Semiconductors (ITRS) has recommended that the maximum value for the EMI is 0.3 mG [5]. To achieve less EMI, there are two important systems: passive canceling systems [6–9] and active magnetic field canceling systems [10]. Usually, the passive-shielding method is the best methodology for magnetic-field reduction

in a nano fab. This paper focuses on creating a model to reduce the EMI for nano fab industries.

With the rapid development of modern nano fabs, EMI has become a critical issue in setting up next-generation semiconductor foundries. It has been reported that when the CMOS progress shows a resolution of fewer than 14 nm, the ELF magnetic fields show significant impacts on semiconductor equipment such as SEMs, TEMs, STEMs, FIB writers, and E-beam writers. Therefore, the recent standard of SEMI [11] suggests that the environment's magnetic field should be maintained below 0.5 mG to prevent malfunction in such sensitive equipment.

To achieve this objective, there are two different systems: the passive [12–14] and active [15] magnetic field canceling systems. In general, passive shielding is the most straightforward method for magnetic field mitigation. However, in the ELF range, shielding using ferrite composite materials significantly increases the foundry fabrication cost. Active magnetic field canceling systems provide an alternative option to alleviate the detrimental effects of ELF magnetic fields. Active canceling systems build up reverse magnetic fields whose intensity and phase are respectively equal to and inverted concerning the environmental field for real-time canceling.

Energy electronic devices are very flexible and capable of handling overpower as high as 10 kW; additionally, these devices are capable of working at frequencies in the range of loads of kHz, with the control being only on the gate terminal of the devices, which makes them easily controllable [16,17]. As technology has grown rapidly, the exposure to EMI has become a new phenomenon in the 20th century with every human being exposed to weak electric and magnetic fields at their workplace and even at home [18]. All electronic devices and power transmission lines produce low-frequency fields in the order of 50 or 60 Hz which have two components: the electric field and the magnetic field [19,20]. Devices such as smartphones and televisions emit these fields with a high-frequency range from 300 MHz–300 GHz, which can be harmful to living organisms [21,22].

The radiation from these devices has a thermal effect, which raises the temperature of body tissues, damaging organs and body cells. According to the International Agency for Research on Cancer (IARC) 2002, low-frequency magnetic fields are carcinogenic [23–32]. In 2011, the World Health Organization (WHO) and the IARC jointly stated that electric and magnetic fields increase the risk of developing brain tumors [33]. When a low-frequency electric field acts on conductive materials, it increases the electric charge on the surface and current flows from the body to the ground. This phenomenon may be related to the human nerve, as the nerves transmit signals by transmitting electrical impulses [34].

Typically, EMI difficulties arise due to unexpected adjustments to voltage (dv/dt) or current (di/dt) levels in the waveform. In a diode rectifier, the line peak can generate a pulse of fast time, and the diode restore current pulse can produce temporary voltage spikes in the line inductor. A dv/dt wave-carrying conductor, which acts as an antenna and a sensitive-signal circuit, looks like noise. EMI problems create communication line interference with sensitive-signal electronics [35–37]. This article focuses on designing a prototype analog active magnetic field cancellation system for nano fab. The experimental results indicate that at 60 Hz, the magnetic field can be significantly reduced from 10 mG to less than 1 mG.

## 2. Theoretical Study and Simulation Models

Three-phase electrical power is a commonly used method to generate, transmit, and distribute alternating-current electrical power. This method is similar to the polyphase system, which is a common method of transmitting energy from electrical networks around the world. In this study, we assumed that four alternating currents $I_R$, $I_S$, $I_T$, and $I_N$ flow in four cables of length 2 L had different phase angles, with the phase shift of $2\pi/3$ between each phase, with a relative phase angle between the phases of $-120$ degrees. The magnetic field B at any point was calculated according to the Biot–Savart law:

$$R_i = r_i \hat{r} - z' \hat{z} \tag{1}$$

$$dl' \times R_i = dz'\hat{z} \times (r_i\hat{r} - z'\hat{z}) = r_i dz'\hat{\phi} \tag{2}$$

where index $i$ = S, R, T, and N.

Substitution of (2) into the Biot–Savart law gives:

$$B_i = \int dB_i = \frac{\mu_0 I_i \hat{\phi}_i}{4\pi} \int_{-L}^{L} \frac{r_i dz'}{(z'^2 + r_i^2)^{\frac{3}{2}}} = \frac{\mu_0 I_i L \hat{\phi}_i}{2\pi r_i \sqrt{L^2 + r_i^2}} \tag{3}$$

Then the total magnetic field is:

$$B = \sum_{i=A}^{C} B_i = \sum_{i=A}^{C} \frac{\mu_0 I_i L \hat{\phi}_i}{2\pi r_i \sqrt{L^2 + r_i^2}} \tag{4}$$

In general, three-phase systems are more economical, prompting many nano fabs to use three-phase power systems [38]. A three-phase power system provides greater power density than a single-phase system at the same voltage, making it a less-expensive process with a smaller wiring size. It is also easier to reduce harmonic currents and load balancing in three-phase systems. These systems are used for electric motors and high-power tools because they optimize the use of electrical capacity to increase power efficiency. The phase angle between each phase is not always zero, and depends on the load type [39].

One-dimensional finite element method (1-D FEM) simulations were performed for the examination of the three-phase, four-cable framework. This method is widely used for solving numerical differential equations arising in engineering and mathematical modeling. Conventional fields of structural analysis, heat transfer, fluid flow, mass transport, and electromagnetic potential are among the typical problem areas in 1-D FEM simulations [40–42]. The cable assemblies were arranged into two types in our reproduction: a counterclockwise framework and a clockwise framework. These arrangements are shown in Figure 1, where R, S, T, and N represent the four alternating current cables with different phase angles, with a phase shift of $2\pi/3$ between each phase, and a relative phase angle between phases of −120 degrees. The reason for this phase change is spatial displacement. The clockwise and counterclockwise framework arrangements help to increase understanding of the paper since the ideal permutation consists of the combination of these frameworks. Figure 1 shows how the optimal ideal permutation system is applied in a nano fab with a three-phase, four-cable system.

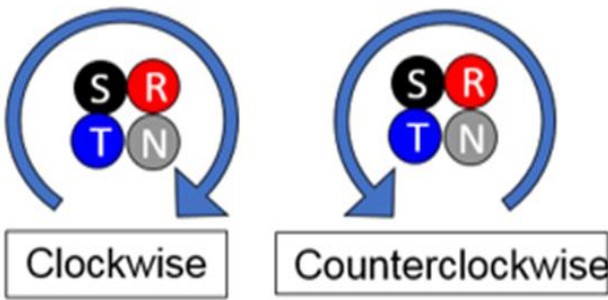

**Figure 1.** Diagram showing clockwise and counterclockwise cable system arrangement.

Three-phase electric force is the basic process used to age, transmit, and disperse the rotary flux electric force. This strategy is similar to the polyphase framework, which is the basic technology for transmitting energy over a global power network [33]. Figure 2 shows a list of the most optimal and ideal permutation cases applicable in the nano fab according to our simulation results. In this study, the three-phase electric power cables (R, S, T, and N) are bound in a square shape as one group. A twelve-group system was simulated in this study, with different phase angles, with the phase shift of $2\pi/3$ between each phase. The relative phase angle between the phases was −120 degrees for the optimization of electromagnetic influence in a high-technology nano fab.

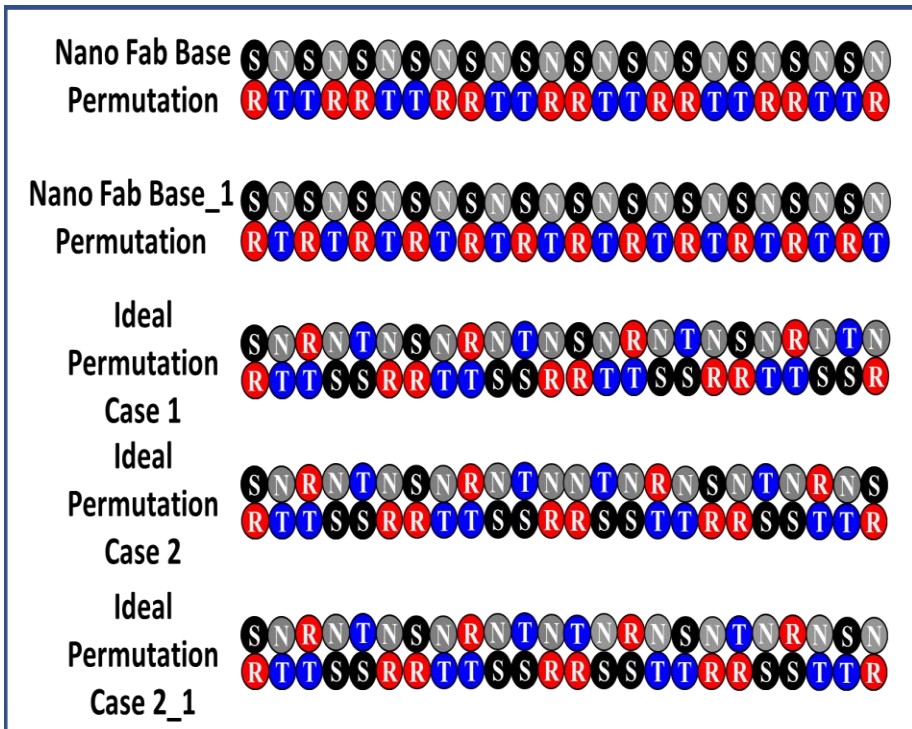

**Figure 2.** The optimal nano fab base permutations (nano fab base permutation and nano fab base_1 permutation) and ideal permutations (ideal permutation case 1, ideal permutation case 2, and ideal permutation case2_1) applied to a nano fab with a three-phase, four-cable (R, S, T, and N) system.

Figures 3 and 4 show the optimal permutation system of the nano fab and ideal power cable permutation system with a three-phase, four-cable (R, S, T, and N) system. The conditions of the simulated atmosphere assume that the current I = 150 A and the frequency f = 60 Hz go to the Z-axis with a length of 2.3 m to 3 m. In our work, the distribution of the magnetic field is focused on the X–Y plane. These high currents applied to the wired lines correspond to the significant power demand in the nano fab. In this article, we present the best permutation with the 1-D system in the nano fab. The results showed a lower EMI with an ideal diffusion method in comparison to the nano fab permutation case.

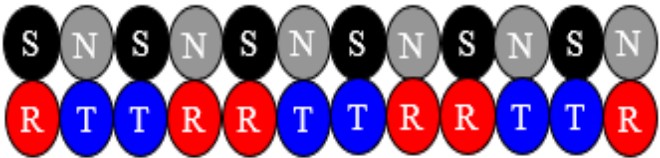

**Figure 3.** Diagram showing the optimal permutation system by nano fab for magnetic field measurement with three-phase, four-cable (R, S, T, and N) system.

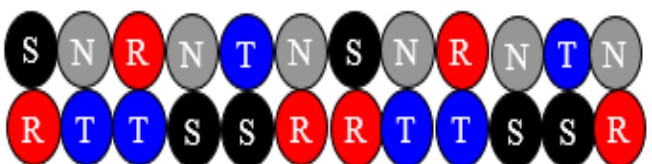

**Figure 4.** Diagram showing the ideal power cable permutation system for magnetic field measurement with three-phase, four-cable (R, S, T, and N) system.

## 3. Results and Discussion

### 3.1. Experimental Results

Figure 5 shows the magnetic field distribution in the surroundings of the nano fab where the electromagnetic field values were measured for several cable distances ranging from 0.1 m to 3 m. At 2.3 m height, rectangular movable plate trays made of iron rods were developed to check the electromagnetic fields in the nano fabs. The magnetic fields were measured by the nano fab by using the permutations with A/m as the units. Maintaining a close proximity to transmission lines and exposure to their electromagnetic interference is harmful to human life, and this EMI may also lead to inaccurate readings in industrial measurement equipment [14–16]. Therefore, this study explains how to reduce electromagnetic interference at a specified distance from cables using the reported permutation systems.

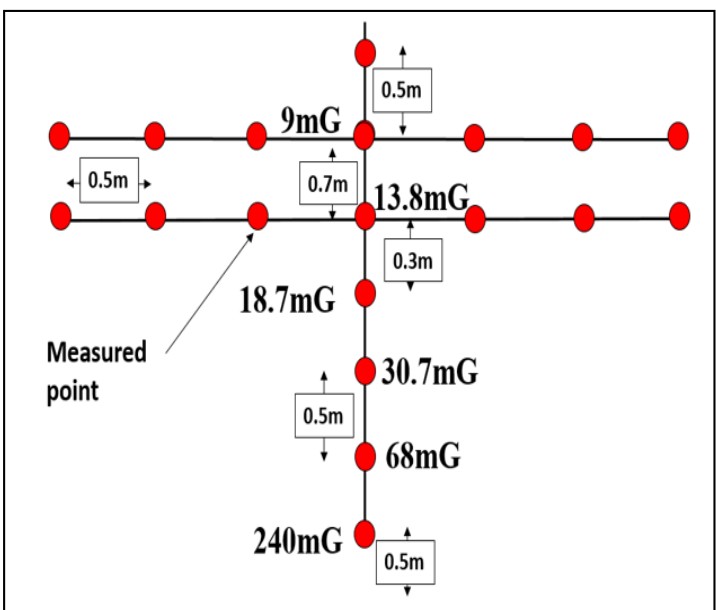

**Figure 5.** Magnetic field distribution of nano fab by using an optimal power cable permutation system with a three-phase, four-cable (R, S, T, and N) system (as shown in Figure 3).

The maximum magnetic field distributions were observed at a radius of 1.8 m with a value of 240 mG. In the closer range of a 0.7 m radius, the magnetic fields were under 18.7 mG. At a 3 m distance from the nano fab, the magnetic field was measured to be 13.8 mG. The average current level for the nano fab base permutation was 154.2 A with a three-phase, four-cable system.

Figure 6 shows the magnetic field distribution in the surroundings of the nano fab in the ideal power cable permutation system. The result can be observed at 3 m from the nano fab. The maximum magnetic field distribution using ideal permutation was observed at a radius of 0.5 m with a value of 18.5 mG. At a 3 m distance, the nano fab permutation had a value of 13.8 mG, whereas in the ideal permutation system it was 0.66 mG. The average current level for an ideal permutation system is 162.5 A with a three-phase, four-cable system.

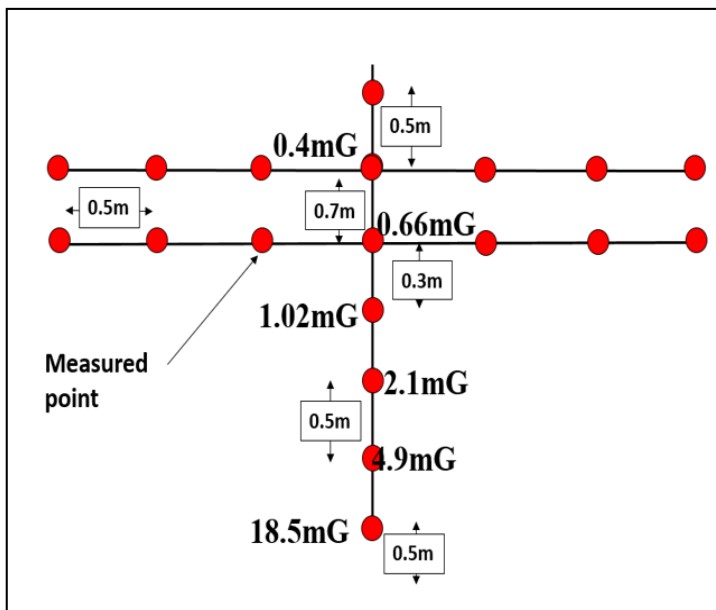

**Figure 6.** Magnetic field distribution of nano fab using an ideal power cable permutation system with a three-phase, four-cable (R, S, T, and N) system (as shown in Figure 4).

### 3.2. Simulated Results

Figure 7 shows the magnetic field distribution at position X = 0 m for all the nano fab-based permutations and ideal permutation cases. At X = 0 m and Y = 0 m, one can observe a very high magnetic field in the range of −5 m to +5 m along the Y-axis. Ideal permutation case 1 shows a lower magnetic field along the Y-axis than any other permutation setup. The results were obtained without any shielding system.

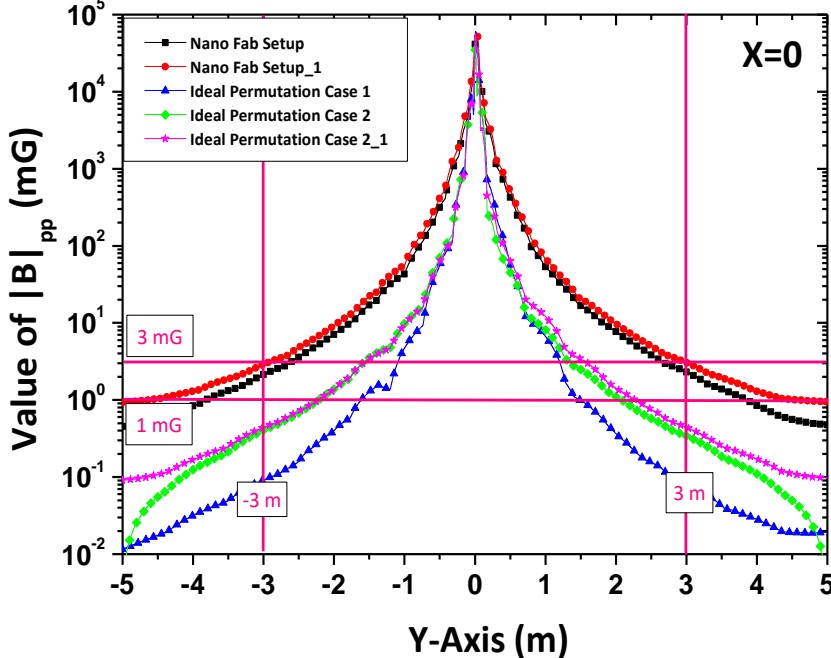

**Figure 7.** The magnetic field at X = 0 with simulated existing case and prediction of the best 1-D permutation results.

Figures 8 and 9 show the magnetic field distribution of the nano fab through the simulation results. The conditions of the simulation environment were current I = 15 A and 750 A at X = 0 m. We chose these values to ensure that the ideal permutation showed

the best results throughout the system with the minimum error rate possible. The results of both systems were obtained without considering any shielding effect. In Figures 8 and 9, the magnetic field distribution for the nano fab-based permutation shows better results than the ideal permutation system at X = 0 m and until Y = 0.25 m.

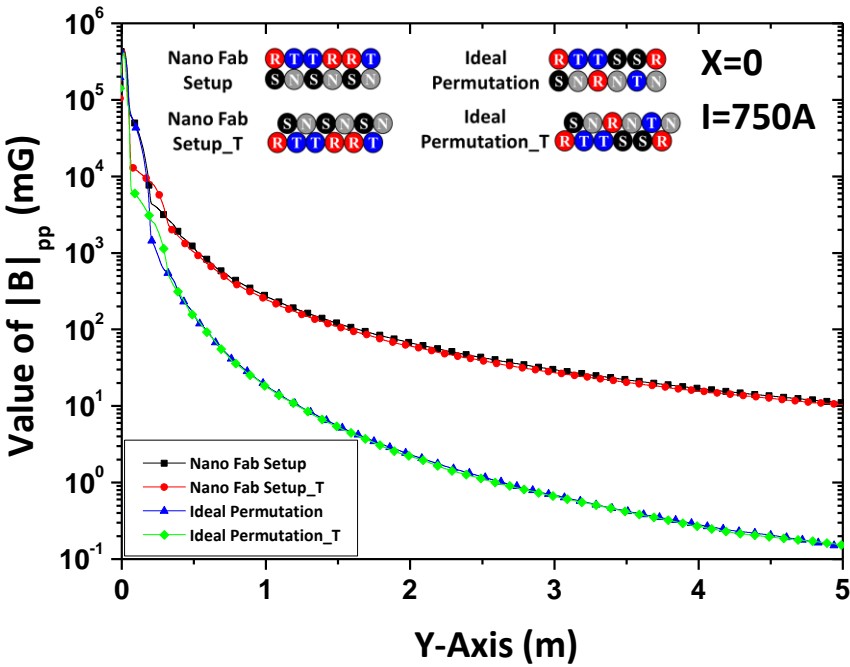

**Figure 8.** Magnetic field distribution of nano fab at position X = 0 m and current level I = 750 A.

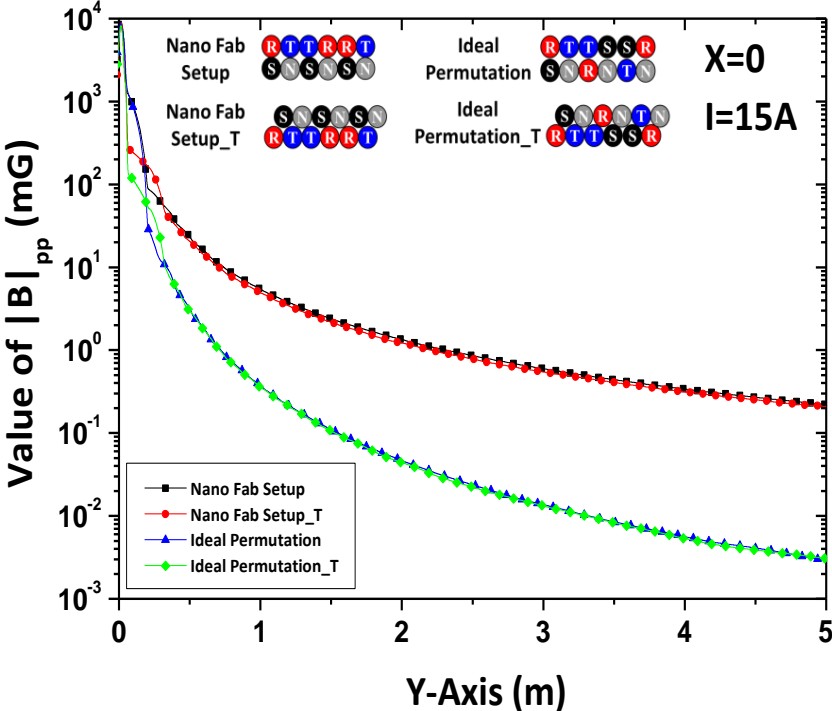

**Figure 9.** Magnetic field distribution of nano fab at position X = 0 m and current level I = 15 A.

The magnetic field values shown in Figures 10 and 11 were measured at Y = 2.3 m and Y = 3 m (Y is the distance from the cable tray and inside cable trap width) by varying the X-axis. At Y = 2.3 m, the nano fab base permutation setup showed a magnetic field

value of 5.1 mG~6.3 mG whereas the ideal permutation system showed the value of 0.19 mG~0.21 mG. The results indicate the effective mitigation of the magnetic field at 2.3 m floor range from the cable trap for the ideal permutation case.

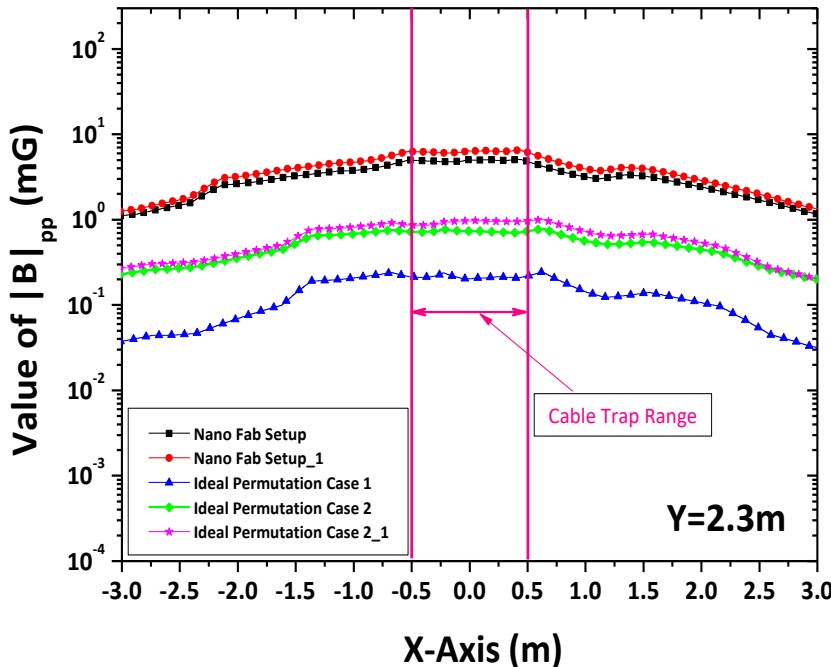

**Figure 10.** The magnetic field at Y = 2.3 m with simulated existing case and predicted best 1-D permutation results.

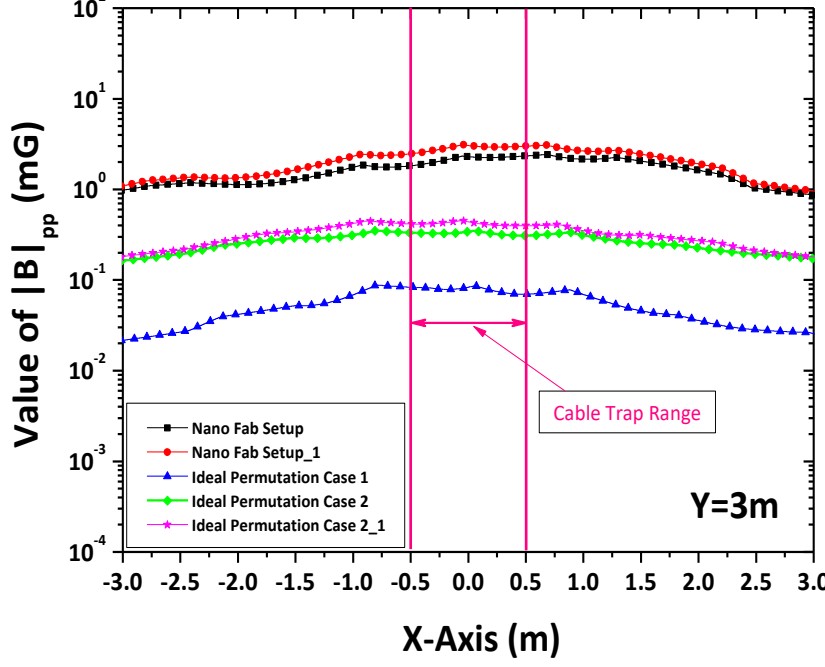

**Figure 11.** The magnetic field at Y = 3 m with simulated existing case and predicted best 1-D permutation results.

*3.3. Discussion*

Earlier in this work, it was necessary to build a model of a high-technology nano fab and simulate this case system by following several steps. First, the magnetic field values

database in a nano fab environment was measured. Then, we simulated serial systematic cases to determine which one matched with the database as above. Next, the best case with the lowest magnetic field values was selected. Finally, the database was compared with the best case to predict the new cable system applied in a high-technology nano fab.

The results of this study indicate that the ideal permutation technique can help to reduce the EMI of the nano fab. From the above trend, Figure 12 shows the comparison of the ideal permutation and optimal permutation in both by simulation and measurement data. Simulation results were obtained using the commercial MagNet (Infolytica Corp., Montreal, QC, Canada) for magnetic mode analysis of the three-phase electric power system. The nano fab measurement results were obtained using the nano fab base permutation. As can be seen in Figures 12 and 13, the ideal permutation method showed much better magnetic field results compared to the nano fab-based permutation. The simulation results were confirmed with the measurement data of ideal permutation and nano fab-based permutation, which showed significant improvement in reducing the magnetic field distribution along the *Y*-axis. The ideal permutation case showed the best electromagnetic interference along the *Y*-axis.

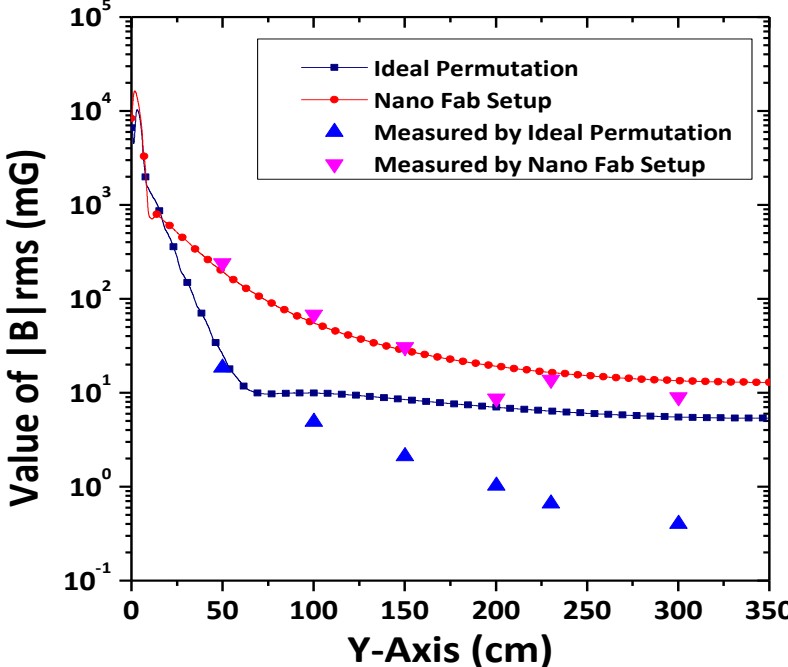

**Figure 12.** Magnetic field comparison for optimal permutation system by nano fab and ideal power cable permutation with a three-phase, four-cable (R, S, T, and N) system.

In this case, the EMI is the electrical noise induced in the wiring by nearby power lines. Electromagnetic interference can interfere with signal transmission. This can have an impact on machines in the manufacturing industry, causing them to malfunction and lose data. Figure 14 shows the implementation of a three-phase, four-cable permutation system in a nano fab. Based on the above-simulated data and real data, the ideal permutation system gave better results in reducing the magnetic field distribution and helped to minimize the effects of high magnetic fields in nano fab environments. Figure 15 shows the practical implementation of the ideal permutation system and the ways to measure EMI at different locations from 0.5 m to 3 m to the ground (near the cable tray system).

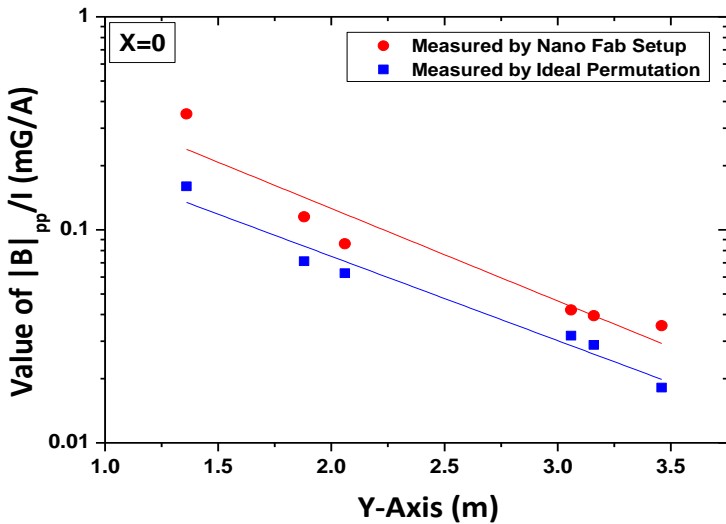

**Figure 13.** Magnetic field comparison of optimal permutation system by nano fab and ideal power cable permutation with a three-phase, four-cable (R, S, T, and N) system at X = 0 m.

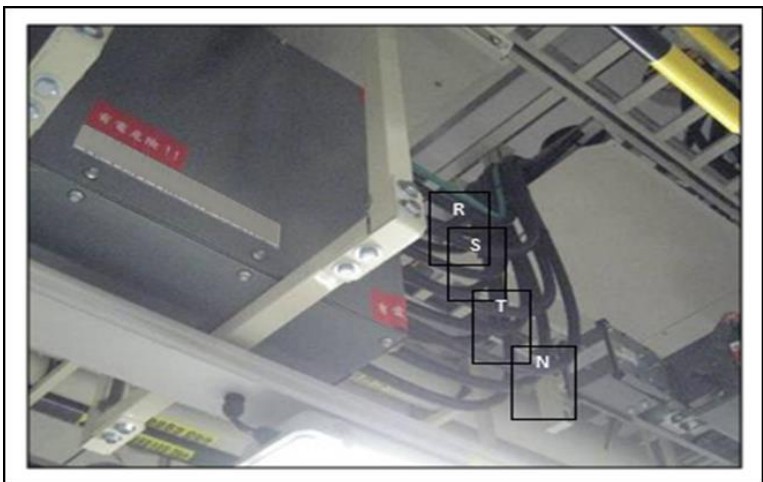

**Figure 14.** Installation of the ideal power cable permutation system with a three-phase, four-cable (R, S, T, and N) system in a nano fab.

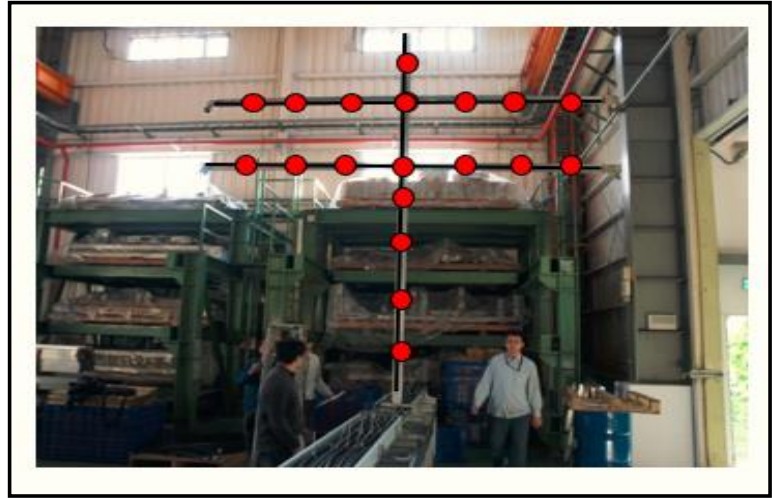

**Figure 15.** Execution and magnetic field measurement point in nano fab using the ideal power cable permutation system with a three-phase, four-cable (R, S, T, and N) system.

## 4. Conclusions

Based on the theoretical research and arithmetic simulation by the ideal permutation method, we observed that moderation of EMI with the ideal permutation of a three-phase, four-cable framework for a high-technology nano fab without any shielding system depressed the magnetic field from 9 mG to 0.4 mG at 3 m range. The simulated magnetic field at 3 m demonstrated an improvement of more than 90% when compared to the nano fab data. The simulated results were clearly in agreement with the data provided by the nano fab. The improvement in the magnetic field boosts the confidence of building new nano fab systems. Based on literature reviews, a reduction in EMI can improve the human body's communication and noise cancellation capabilities. Furthermore, this progressive technique will help to improve the efficiency and cost-effectiveness of nano fabs.

Through the theoretical study and numerical simulation, we predicted the best permutation for reducing EMI noise from a three-phase electric power line system without any shielding system down to 1.2 mG at a 3.0 m distance, applying I = 150 A at 60 Hz with a 12-series cable tray system. Following the above discussions, this study indicated new perspectives that build on the understanding obtained from previous research. The measured magnetic field values of the nano fab are in good agreement with these simulation results. These good results increase confidence in the building of new nano fab systems. Furthermore, this innovation will reduce the cost of EMI shielding in high-technology nano fabs, especially for 40 nm (and below) processes.

**Author Contributions:** Conceptualization, Y.-L.S. and L.-M.C.; investigation, Y.-L.S. and H.-Y.L.; methodology, Y.-L.S. and L.-M.C.; validation, H.-Y.L.; writing—review and editing, Y.-L.S. and L.-M.C.; software, M.K.R.; data curation, M.K.R. and H.-Y.L.; writing—original draft, M.K.R.; funding acquisition, Y.-L.S. All authors have read and agreed to the published version of the manuscript.

**Funding:** This work was supported in part by the Ministry of Science and Technology, Taiwan, through grants MOST 109-2221-E-002-023, MOST 109-2811-E-002-556, MOST 110-2221-E-002-165, MOST 110-2811-E-002-561 and MOST 110-2622-E-468-002.

**Institutional Review Board Statement:** Not applicable.

**Informed Consent Statement:** Not applicable.

**Data Availability Statement:** The study did not report any data.

**Conflicts of Interest:** The authors declare no conflict of interest.

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
