# Peer review of "Control of EMI in High-Technology Nano Fab by Exploitation Power Transmission Method with Ideal Permutation"

_applsci, doi:10.3390/app112411984_

Round 1
Reviewer 1 Report
At the beginning I would like to congratulate the authors on the interesting results. The aim of this article is optimal permutation of
power transmission lines to reduce electro-magnetic influence at high technology nano-Fab. On the basis of the conducted analyzes, correct conclusions were formulated. the article has been improved compared to the previous version while I have such comments. Please explain Figure 14.
Reviewer 2 Report
The manuscript presents a method to reduce the EMI of high-power cables through an ideal permutation.
The manuscript includes a theoretical analysis of the problem and simulations based on Finite Element Method and the comparison to experimental results.
The paper is well written and it explains the research activity performed by the Authors.
Some minor suggestions:
- Line 65: 10KW should be written as 10 kW
- Line 74: 300GHz should be written 300 GHz
- Line 134: cabele should be substituted with cables
- Referring to Figure 12, it should be explained more in detail the difference between measured and simulated ideal permutation.
- Figure 14 is not intelligible.
In conclusion, the manuscript is suitable for the publication in this journal after minor revisions.
Author Response
Please see the attachment

This manuscript is a resubmission of an earlier submission. The following is a list of the peer review reports and author responses from that submission.
Round 1
Reviewer 1 Report
At the beginning I would like to congratulate the authors on the interesting results. The aim of this article is optimal permutation of
power transmission lines to reduce electro-magnetic influence at high technology nano-Fab. On the basis of the conducted analyzes, correct conclusions were formulated. I have a few work-related suggestions:
1. Line 110, Page 3 , I suggests an explanation of the FEM abbreviationLine 149-153,
2. Page 5, I suggest you pay attention to the text formatting as it is different from other text.
3. I suggest that you conduct a literature review of contemporary items. the mentioned literature is quite old. Current reports are missing.
4. The authors refer to the harmfulness of the electromagnetic field. In my opinion, the electromagnetic field limits for the lines are missing and compared with the obtained results. Please comment.
Reviewer 2 Report
Title : “Control of EMI in High Technology Nano-Fab by Exploitation Power Transmission Method with Ideal Permutation”
In this work the authors present an optimal permutation of power transmission lines to reduce electro-magnetic influence at high technology nano-Fab. In this study, the magnetic field was lessened by the mirror array power cable system, and a simulation of results predicted the best permutations to decrease electromagnetic interference (EMI) value below 0.4 mG at working space without any shielding.
General comment: Although the aim of this work is interesting, this manuscript should be reworked to enhance its quality and impact. In particular, the authors should better explain what is the “Ideal of permutation system”
Some detailed comments:
Section “2. Theoretical Study and Simulation Models”
*) The authors should add theoretical and simulation studies, which seem to be not present within the current version of this section. Are there some important formulas to recall ? Please insert this information for interested readers.
Section: 3. Results and Discussion
*) I suggest to split into two different sections this paragraph: Results, where all achieved results are presented without comments, Discussion, where all the achieved new results are compared to the current state of the art. Please rework these sections accordingly.
*) All figures should of high resolution and quality. All the captions should be meaningful “per se”: please improve accordingly.
Figure 5. Magnetic field distribution at Nano-Fab by using Nano-Fab permutation
Figure 6. Magnetic field distribution at Nano-Fab by using an ideal power cable permutation system
*) These figures are not totally clear. Please improve the captions and the explanation. Where are the claimed “simulations”. Could the authors show the value of the magnetic field in all the points through FE simulations ? Please rework accordingly.
Lines: “From this study, the ideal permutation technique helps in reducing the EMI of Nano Fab. 209
From the above trend curve, Fig.12 and Fig.13 clearly show that there is a huge improvement in reducing the magnetic field distribution along Y-axis. The ideal permuta- 211
tion case 1 shows the best electromagnetic interference along Y-axis”
*) The authors should rework these point, better explaining what the “permutation technique” is.
Figure 14. Implantation of three-phase four cables (R, S, T, N) system in Nano Fab
Figure 15. Execution of Ideal permutation system and measuring points of EMI in Nano Fab
*) These pictures are interesting but it is not clear their values. Please provide high resolution and important figures to support this work.
Round 2
Reviewer 2 Report
Title : “Control of EMI in High Technology Nano-Fab by Exploitation Power Transmission Method with Ideal Permutation”
In this work the authors present an optimal permutation of power transmission lines to reduce electro-magnetic influence at high technology nano-Fab. In this study, the magnetic field was lessened by the mirror array power cable system, and a simulation of results predicted the best permutations to decrease electromagnetic interference (EMI) value below 0.4 mG at working space without any shielding.
General comment: This work has been partially revised. However, some further work should be added to improve the quality and the impact of this manuscript. In particular, some points of this work are still not clear and should be clarified to all the interested readers.
Some detailed comments:
*) The section “2. Theoretical Study and Simulation Models” should be reworked. The authors should add theoretical and simulation studies. In addition, all figures should be clearly described within the main text.
Figure 1.
*) Please clarify further this Figure with a suitable caption. Also all symbols used should be clearly explained.
Figure 2
this figure is not clear. Please provide a suitable caption. In particular, it is not clear how many wires should be considered and why.
Lines: “1-D FEM (Finite Element Method) simulations are performed for the examination of the 110
three-phase four cable framework with reading data equipment. This method is widely 111
for numerically solving differential equations arising in engineering and mathematical 112
modeling. Conventional fields of structural analysis, heat transfer, fluid flow, mass 113
transport, and electromagnetic potential are among the typical problem areas [41-43].
*) the authors should explain in all details the “1D FEM simulations”.
*) Figure 3 and 4 are not clear. Please explain.
*)Figure 5. Magnetic field distribution at Nano-Fab by using Nano-Fab permutation
*) Figure 6. Magnetic field distribution at Nano-Fab by using an ideal power cable permutation system
*) These figures are not totally clear. Please improve the captions and the explanation. Where are the claimed “simulations”. Could the authors show the value of the magnetic field in all the points through FE simulations ? Please rework accordingly.
Lines: Figure 12. Comparison of nano-fab base permutation system magnetic field with ideal permutation system.
*) Please explain in a better way what is the permutation system magnetic field ?
Figure 14. Implantation of three-phase four cables (R, S, T, N) system in Nano Fab
Figure 15. Execution of Ideal permutation system and measuring points of EMI in Nano Fab
*) These pictures are interesting but it is not clear their values. Please provide high resolution and important figures to support this work.
Round 3
Reviewer 2 Report
Title : “Control of EMI in High Technology Nano-Fab by Exploitation Power Transmission Method with Ideal Permutation”
In this work the authors present an optimal permutation of power transmission lines to reduce electro-magnetic influence at high technology nano-Fab. In this study, the magnetic field was lessened by the mirror array power cable system, and a simulation of results predicted the best permutations to decrease electromagnetic interference (EMI) value below 0.4 mG at working space without any shielding.
General comment: The work has been only partially revised. The main changes should be still performed within the main text. Indeed, even if the authors provide a rebuttal letter where they explain the corrections, the novel version should contain all the reworked parts within the main text. In particular, the authors could refer to the previous review to check all the needed changes to do within the novel version of the manuscript.
Some detailed comment:
*) 2. Theoretical Study and Simulation Models
The mathematical explanation should be referred to the studied system. Thus, please provide a detailed scheme of the studied system. What is r with reference to the figure 15 ?
Lines “-D FEM (Finite Element Method) simulations are performed for the examination of the
118 three-phase four cable framework. This method is widely for solving numerical differ-
119 ential equations arising in engineering and mathematical modelling. Conventional fields
120 of structural analysis, heat transfer, fluid flow, mass transport, and electromagnetic po-
121 tential are among the typical problem areas in 1-D FEM simulations [41-43]. The cable
122 assembling’s were arranged into two sorts which were categorized as the counterclock-
123 wise framework and clockwise framework in our reproduction, these arrangements are
124 shown in Figure 1. Where, S, R, T, and N are the four alternating current cables with dif-
125 ferent phase angles and with the phase shift of 2π/3 between each phase and the relative
126 phase angle between the phases is -120 degrees. The reason for this phase change is spa-
127 tial displacement. The understanding of the clockwise and counterclockwise framework
128 arrangements helps in a better understanding of the paper since the ideal permutation
129 consists of the combination of these frameworks. From Figure 1 we can see how the op-
130 timal ideal permutation system applies in nano-fab with a three-phase four-wire system “
*)The authors should provide a detailed description of the meaning of 1D FEM simulations just in this studied case.
Figure 2. The optimal nano-fab base permutations and ideal permutations applied to fab with three
134 phase four cabele (S, R, T, and N) system.
*) please rework this figure and provide a suitable and fully descriptive caption. The order and the number of cable is not clear, please explain.
Figure 3. Diagram showing the optimal permutation system by nano-fab with three phase four
146 cable (S, R, T, and N) system
147 .
148 Figure 4. Diagram showing the ideal power cable permutation system with three phase four cable
149 (S, R, T, and N) system.
* ) The captions of these figure should be made more clear and fully descriptive. A comparison with figure 15 where the real cable disposition is shown is needed to better understand the permutations.
3.1. Experimental Results
Figures 5 and 6.
*) The authors should enhance the quality of the captions (as explained in the previous review)
3.2. Simulated Results
*) See the previous review. In general all the figures should be in high resolution with fully explanatory captions.
Figure 5. Magnetic field distribution at Nano-Fab by using Nano-Fab
permutation
Figure 6. Magnetic field distribution at Nano-Fab by using an ideal power cable
permutation system
These figures are not totally clear. Please improve the captions and the
explanation. Where are the claimed “simulations”. Could the authors show the
value of the magnetic field in all the points through FE simulations ? Please
rework accordingly
Figure 14. Implantation of three-phase four cables (R, S, T, N) system in
Nano Fab
Figure 15. Execution of Ideal permutation system and measuring points of EMI
in Nano Fab
*) These pictures are interesting but it is not clear their values. Please provide
high resolution and important figures to support this work.
"Some of the key figures can’t the presented in the manuscript due to mutual agreement with the fab. The authors wish the reviewer will understands the agreement policies."
*) Please, present all the needed figures.
Round 4
Reviewer 2 Report
Title : “Control of EMI in High Technology Nano-Fab by Exploitation Power Transmission Method with Ideal Permutation”
In this work the authors present an optimal permutation of power transmission lines to reduce electro-magnetic influence at high technology nano-Fab. In this study, the magnetic field was lessened by the mirror array power cable system, and a simulation of results predicted the best permutations to decrease electromagnetic interference (EMI) value below 0.4 mG at working space without any shielding.
General comment: Although the aim of this work seem to be interesting, this manuscript should be still reworked to enhance its quality and impact. Unfortunately, the authors should better explain not only the previous underlined points (as detailed in the previous round of revisions e.g. what the “Ideal of permutation system” is, etc ) and better present the theoretical framework used to provide their results (i.e., a clear explanation of how the permutation theory has been used and implemented in this work), but also the new added contribution (in yellow in the main text) should be deeply reworked to provide a sufficient quality for a standard scientific contribution.
Some detailed comments:
Lines: “1-D FEM (Finite Element Method) simulations are performed for the examination of the 116
three-phase four cable framework. This method is widely for solving numerical differen- 117
tial equations arising in engineering and mathematical modelling. Conventional fields of 118
structural analysis, heat transfer, fluid flow, mass transport, and electromagnetic potential 119
are among the typical problem areas in 1-D FEM simulations [40-42]. The cable assem- 120
bling’s were arranged into two sorts which were categorized as the counterclockwise 121
framework and clockwise framework in our reproduction, these arrangements are shown 122
in Figure 1. Where, R, S, T, and N are the four alternating current cables with different 123
phase angles and with the phase shift of 2π/3 between each phase and the relative phase 124
angle between the phases is -120 degrees. The reason for this phase change is spatial dis- 125
placement. The understanding of the clockwise and counterclockwise framework ar- 126
rangements helps in a better understanding of the paper since the ideal permutation con- 127
sists of the combination of these frameworks. From Figure 1 we can see how the optimal 128
ideal permutation system applies in nano-fab with a three-phase four-wire system. 12 2
*) The authors should explain in details how they implemented 1-D FEM in this work. Please rework accordingly.
Lines. “Figure 2. The optimal nano-fab base permutations (nano-fab base permutation and nano-fab 132
base_1 permutation) and ideal permutations (ideal permutation case 1, ideal permutation case 2, 133
and ideal permutation case2_1) applied to fab with three phase four cabele (R, S, T, and N) system.
“
*) Here it is not clear what is the number of the cable to provide the best permutation. Please explain in a detailed way, and change the figure accordingly.
Lines: “3.3. Discussion 220
Earlier in this work, we must build a model in high technology nano-fab and simulate this 221
case system by following several steps. The first, magnetic field values database in an en- 222
vironment of nano-fab was measured. And then simulating serial systematic cases found 223
which one was matching with the database as above. Next, from the second step got the 224
best one which was taken on the lowest magnetic field values. Finally, compare the data- 225
base with the best case to predict the new cable system applying in high technology nano- 226
fab. 22
“
*) The discussion section should be reworked. A suitable discussion should be provided to better underline the value of this work and to compare the new results to the state of the art. The quality of the language should be necessarily enhanced to allow the interested readers to follow the meaning of the text.
Lines: “Figure 14 and Figure 15 shows the implementation of a three-phase 241
four-wire permutation system on a nano-fab. Based on the above-simulated data and real 242
data, the ideal permutation system gives better results in reducing the magnetic field dis- 243
tribution and helps us to minimize the effects of the high magnetic field in nano-fab envi- 244
ronments. Figure 16 shows the practical implementation of the ideal permutation system 245
and the ways to measure EMI at different locations from 0.5m to 3m to the ground (near 246
the cable tray system). “
*) The meaning of these lines is not clear. Please rework them accordingly.
Lines: “Figure 14. Optimal permutation system by nano-fab for magnetic field measurement with three 249 phase four cable (R, S, T, and N) system “
*) This figure is absolutely not clear and the meaning is totally not cleat. Perhaps this is an error.. ? Please replace this figure with another one more meaningful.
Please provide also more meaningful captions of the remaining figures